# Hemorrhagic Resuscitation Guided by Viscoelastography in Far-Forward Combat and Austere Civilian Environments: Goal-Directed Whole-Blood and Blood-Component Therapy Far from the Trauma Center

**DOI:** 10.3390/jcm11020356

**Published:** 2022-01-12

**Authors:** James H. Lantry, Phillip Mason, Matthew G. Logsdon, Connor M. Bunch, Ethan E. Peck, Ernest E. Moore, Hunter B. Moore, Matthew D. Neal, Scott G. Thomas, Rashid Z. Khan, Laura Gillespie, Charles Florance, Josh Korzan, Fletcher R. Preuss, Dan Mason, Tarek Saleh, Mathew K. Marsee, Stefani Vande Lune, Qamarnisa Ayoub, Dietmar Fries, Mark M. Walsh

**Affiliations:** 1Department of Medicine Critical Care Services, Inova Fairfax Medical Campus, Falls Church, VA 22042, USA; jlantrymd@gmail.com; 2Department of Critical Care Medicine, San Antonio Military Medical Center, Fort Sam Houston, San Antonio, TX 78234, USA; phillipmason@yahoo.com; 3Department of Emergency Medicine, Indiana University School of Medicine—South Bend, Notre Dame, IN 46617, USA; mglogsdo@iu.edu (M.G.L.); cmbunch@iu.edu (C.M.B.); 4Department of Emergency Medicine, St. Joseph Regional Medical Center, Mishawaka, IN 46545, USA; ethan.peck@valpo.edu (E.E.P.); cfloranc@alumni.nd.edu (C.F.); korzanjl@gmail.com (J.K.); 5Department of Surgery, Ernest E. Moore Shock Trauma Center at Denver Health and University of Colorado Health Sciences Center, Denver, CO 80204, USA; Ernest.moore@dhha.org (E.E.M.); hunter.moore@ucdenver.edu (H.B.M.); 6Pittsburgh Trauma Research Center, University of Pittsburgh Medical Center, Pittsburgh, PA 15213, USA; nealm2@upmc.edu; 7Department of Trauma Surgery, Memorial Leighton Trauma Center, Beacon Health System, South Bend, IN 46601, USA; sthomas@beaconhealthsystem.org; 8Department of Hematology, Michiana Hematology Oncology, Mishawaka, IN 46545, USA; rkhan@mhopc.com; 9Department of Quality Assurance and Performance Improvement, St. Joseph Regional Medical Center, Mishawaka, IN 46545, USA; gillesla@sjrmc.com; 10Department of Orthopaedic Surgery, UCLA Santa Monica Medical Center and Orthopaedic Institute, Santa Monica, CA 90404, USA; fletcher.preuss@gmail.com; 11Department of Medical Science and Devices, Haemonetics Corporation, Braintree, MA 02184, USA; dan.mason@haemonetics.com; 12Department of Critical Care Medicine, St. Joseph Regional Medical Center, Mishawaka, IN 46545, USA; Drsaleh20@gmail.com; 13Department of Graduate Medical Education, Naval Medical Center Portsmouth, Portsmouth, VA 23708, USA; mkmarsee@gmail.com; 14Department of Emergency Medicine, Naval Medical Center Portsmouth, Portsmouth, VA 23708, USA; stefani.vandelune@gmail.com; 15Andeshgah Library, Kabul 1006, Afghanistan; qamarayoub16@gmail.com; 16Department of Surgical and General Care Medicine, Medical University of Innsbruck, 6020 Innsbruck, Austria; dietmar.fries@i-med.ac.at

**Keywords:** viscoelastic testing, far forward, austere environment, resuscitation, goal-directed therapy, whole blood, blood-component therapy

## Abstract

Modern approaches to resuscitation seek to bring patient interventions as close as possible to the initial trauma. In recent decades, fresh or cold-stored whole blood has gained widespread support in multiple settings as the best first agent in resuscitation after massive blood loss. However, whole blood is not a panacea, and while current guidelines promote continued resuscitation with fixed ratios of blood products, the debate about the optimal resuscitation strategy—especially in austere or challenging environments—is by no means settled. In this narrative review, we give a brief history of military resuscitation and how whole blood became the mainstay of initial resuscitation. We then outline the principles of viscoelastic hemostatic assays as well as their adoption for providing goal-directed blood-component therapy in trauma centers. After summarizing the nascent research on the strengths and limitations of viscoelastic platforms in challenging environmental conditions, we conclude with our vision of how these platforms can be deployed in far-forward combat and austere civilian environments to maximize survival.

## 1. Introduction

Resuscitation in far-forward combat zones has its roots in the early 19th century based on insights gained during the disastrous French invasion of Russia during the Napoleonic wars. Grande Armée surgeon Dominique-Jean Larrey noted that injured soldiers who were bivouacked nearest to campfires had higher rates of gangrene and death compared to those who remained in colder locations. Larrey concluded that this phenomenon was due to “asphyxia” of the affected limb(s), which could “preserve its life… if the cold was removed by degrees, or if the person affected by it pass into a more elevated temperature by degrees’’ [1]. In Larrey’s time, amputation was the definitive treatment for severe trauma. Thus, hypothermic vasoconstriction was preferred because it resulted in less blood loss and pain. Physiologist Walter Cannon noted a similar phenomenon during his experiences on the Western Front in World War I, but took an opposite view on hypothermia. He believed that patients should be kept warm to maximize perfusion until prompt surgical intervention could definitively control bleeding [2]. Despite having no knowledge of coagulopathy, he was an early proponent of whole-blood (WB) transfusion over saline-based infusions because WB was “the most effective means of dealing with cases of continued low blood pressure, whether due to hemorrhage or shock” [3]. From these and many other insights, Cannon would go on to accurately describe the core mechanisms of hypovolemic shock, many of which are still valid today [2,3].

Despite their delayed publication, Cannon’s views were widely shared during World War I, and the US Army Medical Department quickly adopted citrated WB administration to combat shock. Early in World War II, plasma administration was initially used in far-forward resuscitation techniques when WB could not be obtained. However, by 1945, plasma had completely replaced WB because WB was prioritized for pre-operative stabilization. By the Korean War it was lamented by Dr. Walter L. Bloom, “how quickly the World War II experience seemed to have been forgotten and how the tendency was again evident to concentrate on agents other than WB in the management of combat and other casualties” [4]. The difficulty of sourcing and storing sufficient quantities of WB and blood products, as well as the high risk of hepatitis from transfusions, gradually led to an emphasis on crystalloid infusions to achieve targeted blood pressures in combat resuscitations during the Vietnam War [4]. However, this technique was similarly prone to failure, largely due to the poor understanding of the coagulopathy of traumatic blood loss. Post-Vietnam War, experimental work by G.T. Shires in the late 1960s and early 1970s supported replenishment of interstitial sodium with crystalloid, which promulgated standard ratios of 3:1 of crystalloid to blood products [5,6]. Civilian Advance Trauma Life Support (ATLS) guidelines also incorporated this 3:1 ratio and recommended 2 L crystalloid at initial resuscitation for hemorrhaging patients [4]. However, this recommendation was largely misapplied and resulted in high mortality secondary to rapid dilution of coagulation factors that were already depleted by severe trauma. Thus, far-forward combat methods of resuscitation were primed for a paradigm shift back to balanced blood-component resuscitation [7].

The shift back to WB began in the early 1990s (Figure 1). A sentinel event by the US Armed Forces in Somalia facilitated this transition. On 3 September 1993, a medical team experienced a shortage of blood products while treating Spec. Edward J. Nicholson, a soldier who was the victim of a severe shark attack off the coast of Mogadishu, Somalia. The division immediately organized an emergency WB collection program under the direction of Col. Denver Perkins. Unfortunately, Nicholson did not survive, but the division accumulated a stockpile of 120 units of WB. One month later, during the *Black Hawk Down* incident, this stockpile was critical in early resuscitations at the October 3rd Battle of Mogadishu [4,8]. This experience led the military to initiate an extensive review of resuscitation protocols under Lt. Col. John Holcomb, which culminated with the inclusion of WB use in the Army’s extensive third revision of the *Emergency War Surgery* handbook [9].

Concurrently, a civilian trial of 598 patients with hypotension secondary to penetrating torso trauma showed that delaying fluid therapy until bleeding was surgically controlled reduced overall mortality as well as in-hospital morbidity [10]. This and other work led to a consensus statement in 2003 that directly challenged the role of crystalloid in early trauma resuscitation [11], notably citing Cannon as an early critic of such resuscitation strategies. From 2003 to 2005, during Operation Iraqi Freedom, the military was able to test the strategies that had been developed since Somalia. Borgman et al. found that massive resuscitation of military trauma patients with a 1:1 ratio of fresh frozen plasma (FFP) to red blood cells (RBCs) resulted in a 60% reduction in mortality compared to patients treated with a 1:8 ratio [12]. As the evolution of standard therapies progressed from crystalloid driven protocols to fresh WB in the combat theatres of Iraq and Afghanistan, the ability of forward-deployed medics to effectively collect and administer WB in the austere setting has progressed. Use of WB is now embedded in far-forward military resuscitation strategies for hemorrhaging patients [8,13].

The recognized improvements in outcomes from transfusion protocols that mirrored WB led to a call for increased focus on addressing early coagulopathy in far-forward resuscitation [14]. In 2008, Plotkin et al. published a landmark study that detailed how viscoelastic hemostatic assays (VHAs) had been a better guide to predict coagulopathy and the need for massive transfusion (MT) of injured soldiers in a combat hospital compared to conventional coagulation assays (CCAs) (e.g., aPTT, PT/INR, fibrinogen, and platelet count) [15]. This was the first study demonstrating the feasibility of using point-of-care (POC) VHAs in the far-forward combat environment and suggested that WB protocols may be augmented by individualized and goal-directed blood-component therapy (BCT) guided by VHAs.

For the remainder of this review, “WB” refers to the complete range of WB storage and processing types: warm-fresh WB, cold-fresh WB, and cold-stored (processed) WB. The quality differences among these three storage methods of WB are beyond the scope of this review. The definition of “WB” itself is not uniform and depends on local and institutional standards for storage length and temperature, transfusion-transmitted infection screening, leukoreduction, and antibody titer threshold. Military and civilian austere environments traditionally rely on fresh WB, which is likewise not uniformly defined and largely depends on the local transfusion specialists’ discretion. Warm-fresh WB for trauma resuscitation is variably defined as blood stored at room temperature for a maximum of 24 h. Cold-fresh WB is variably defined as refrigerated blood stored <48 or <72 h. Cold-stored WB is refrigerated >48 h and up to 21 days at some centers [7,16,17,18].

**Figure 1 jcm-11-00356-f001:**
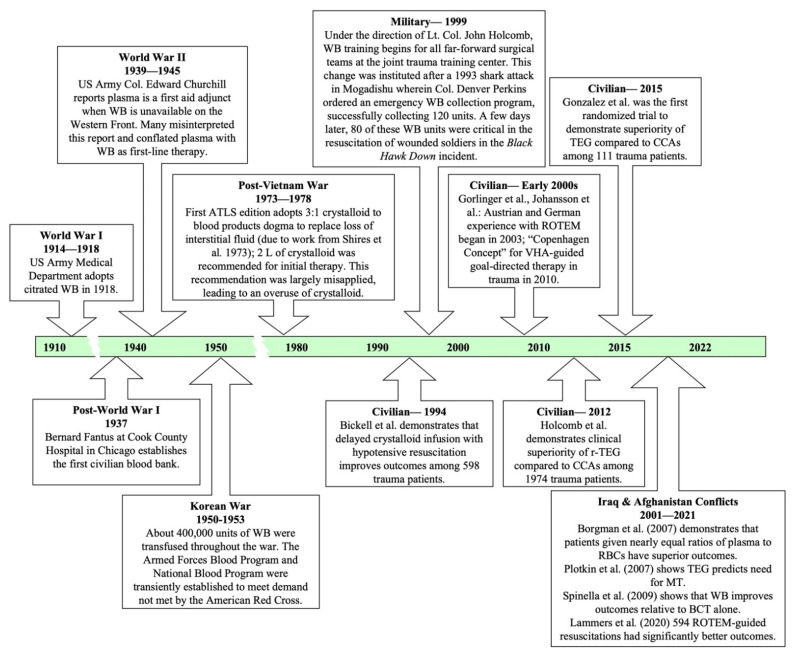
Brief overview of the evolution of trauma resuscitation since the early 1900s in military and civilian settings [4,10,12,15,19,20,21,22,23]. ATLS, Advanced Trauma Life Support; BCT, blood-component therapy; CCAs, conventional coagulation assays; MT, massive transfusion; ROTEM, rotational thromboelastometry; TEG, thromboelastography; r-TEG, rapid thromboelastography.

## 2. Modern Resuscitation in Civilian Environments

In 2011, a worldwide collaboration between civilian and military personnel set out to develop and implement improved protocols for severe trauma in challenging settings. The initiative began as a collaboration between the North Atlantic Treaty Organization (NATO) and the Norwegian Naval Special Operations Commands to establish remote damage control resuscitation (DCR) for a rural prehospital WB program which was organized under the title of Trauma Hemostasis and Oxygenation Research (THOR) Network [24]. Initially, they were focused on straightforward civilian applications of lessons learned in military theaters—such as resuscitation during the transport of trauma patients in rural settings—but over time they expanded their scope to include exotic settings such as civilian maritime cruises where blood products and hospital services are lacking [24]. THOR researchers have also pushed the field of resuscitation philosophically by arguing that the benefits of ubiquitous use of O+ blood in resuscitation vastly outweighs the risks [25]. Numerous scientific contributions from different sources have added to the establishment of WB administration in far-forward combat and civilian austere and urban environments commonly used today [24,25,26,27,28,29,30,31].

In 2020, approximately 24% of all level 1 trauma centers in the United States used WB for resuscitation [32,33]. However, total WB utilization generally ranges from two to six units during prehospital transport [34,35,36]. The largest United States survey of 8494 patients resuscitated for severe hemorrhage has recently confirmed that the number of WB units given was a small fraction of the total number of units that patients received [37]. Specifically, of these 8494 patients, 280 received WB as an adjunct to BCT within the first 24 h of resuscitation. Only 1 unit of WB was given for 95% of the 280 patients and only 5% received 2 units of WB. It is also generally given “blind,” in the sense that little is known about the patient’s hemodynamic or coagulopathic state at the time of WB administration aside from severe ongoing hemorrhage.

VHAs (thromboelastography [TEG^®^; manufactured by Haemonetics Corporation, Boston, MA, USA] and rotational thromboelastometry [ROTEM^®^; manufactured by Werfen, Barcelona, Spain]) allow physicians to augment the optimal standard of care with individualized changes which is analogous to modern approaches to cancer that involve pharmacogenomics to find the individualized drug regimen. Following fixed ratio 1:1:1 guidelines may lead to preventable deaths because trauma-induced coagulopathy (TIC) lies upon a spectrum wherein several factors contribute to the individual patient’s hemostatic derangement (e.g., presence of traumatic brain injury, penetrating or blunt mechanism, initial resuscitation methods, time from injury, genetic hematologic makeup of the patient) [34,35,38,39]. In this light, the question is not whether VHAs should be employed in these settings, but rather how early in the patient’s care VHAs can be used to improve outcomes in patients needing resuscitation. Initial resuscitation with WB ideally occurs within the “golden hour” or even within the “platinum 10 min.” For example, in the military soldiers receive blood products from the “golden hour boxes” in the field as soon as possible after combat-related injuries [40]. Shortly following initial resuscitation with WB, and once more advanced medical support is available (e.g., at a battalion aid station or during air/ground transport), VHAs may play a pivotal role in preventing death.

The Shock, Whole Blood, and Assessment of TBI (SWAT, NCT03402035), Pragmatic, Prehospital, Type O, Whole Blood Early Resuscitation (PPOWER, NCT03477006), and Type O Whole Blood and Assessment of Age During Prehospital Resuscitation (TOWAR, NCT04684719) trials in the United States and the Sang Total pour la Réanimation des Hémorragies Massives (STORHM) trial in France were initiated to definitively study the role of WB [36,41,42]. Importantly, each of these studies will use VHAs to guide goal-directed therapy after initial WB administration.

VHAs are uniquely positioned to provide important information at the time of WB administration and during subsequent resuscitation [43]. Evaluation of trauma patients in combat has revealed that early TIC is associated with mortality; in tandem, the earlier the coagulopathy is addressed, the better the prognosis. There is a developing interest in using VHAs to aid in goal-directed BCT of the bleeding trauma patient as close to the scene of injury as possible, whether that injury occurred in the far-forward combat area, or the civilian austere or urban environments [44,45]. Incidence of TIC has been reported in 24–36% of all trauma admissions to the emergency department, further increasing the importance and usability of POC VHAs in prehospital environments for improved results upon arrival [46,47,48,49,50]. The use of versatile and portable VHAs in the urban environment have been increasingly adopted for the immediate use in trauma resuscitation scenes and in studies concerning prehospital resuscitation [51,52]. VHAs and to some extent CCAs, predict mortality in far-forward combat and civilian austere or urban environments [44,53,54,55,56,57,58,59,60].

The use of VHAs to monitor transfusion requirements has been widely employed in adult and pediatric trauma [61,62], liver transplantation [63], cardiac surgery [64,65], traumatic brain injury [66,67], and postpartum hemorrhage [68]. VHAs have also shown to be cost-neutral compared to CCAs [69]. However, despite VHA’s capability to provide timely information that enables goal-directed therapy, its use outside of operating theaters and similarly controlled settings has generally been limited.

Recently, VHA-guided resuscitation was challenged by the implementing Treatment Algorithms for the Correction of Trauma-Induced Coagulopathy (iTACTIC) trial [70]. This trial demonstrated no significant difference in 28-day mortality between those patients who received VHA-guided care versus standard therapy; yet many confounding variables must be considered. First, there was a relatively low incidence of TIC in both groups despite high injury severity scores. This was reflected by the low percentage of patients who received MT under the traditional definition (defined as ≥10 units RBCs in 24 h). Only 26% of patients in the VHA-guided group and 28% in the standard therapy group received MT at 24 h post-injury. Therefore, one of the study limitations was a patient population predominated by less acutely ill patients. Second, the study used a per-protocol analysis that excluded patients who died within 60 min of CCA or VHA testing. Because the study excluded patients who presumably died within or close to the golden hour, it thereby excludes patients who may have derived the most benefits from early VHA-guided resuscitation because the pathophysiology of TIC is best corrected by intervention within the first few hours of injury [71]. Third, among the seven centers involved in the iTACTIC trial, the authors reported challenges in achieving performance homogeneity with regard to VHA-guided care. In tandem, this raises concern for the validity of the VHA-guided group because there is a learning curve with any new test, such as VHAs, and thresholds for administering BCTs vary based on hospital protocols and VHAs [72,73]. While some centers had years of experience with VHAs in trauma, other centers only began VHA use at the start of the trial. Finally, there was a high incidence of traumatic brain injury which confounds causal links to death by exsanguination [67]. Consequently, VHA-guided BCT for trauma remains controversial.

For the remainder of this review, we shall shift our focus to the two primary VHA platforms, TEG^®^/ROTEM^®^, their capabilities and limitations, and current barriers to their more widespread adoption. We will conclude with specific future applications of VHAs in both far-forward combat and austere civilian environments.

## 3. Thromboelastography (TEG^®^) and Rotational Thromboelastometry (ROTEM^®^)

VHAs were first developed as WB assays based on work performed by Hartert after World War II [74]. VHAs assess the entire coagulation cascade from clot initiation and formation through clot termination including fibrinolysis. While both TEG^®^ and ROTEM^®^ yield similar graphical outputs, their modalities differ in ways that are potentially significant under adverse environmental conditions, which are discussed later in this review. Small technical and nomenclature differences exist between TEG^®^ and ROTEM^®^. For example, maximum clot strength is the surrogate endpoint measurement (maximum amplitude [MA] on TEG^®^ or maximum clot firmness [MCF] on ROTEM^®^). A decreased MA on TEG^®^ (or MCF on ROTEM^®^) may indicate transfusion of platelets or fibrinogen concentrates. A prolonged reaction (R)-time on TEG^®^ (or clotting time [CT] on ROTEM^®^) indicates coagulation factor deficiency/aberrancy and may indicate treatment with plasma, prothrombin complex concentrate, or factor concentrates. Results are generated in minutes and can be interpreted in real time to provide information about the best course of treatment. There are varying transfusion thresholds depending on the specific assay and activators and/or inhibitors of coagulation used, whether in TEG^®^ or ROTEM^®^ [73,75,76,77].

VHAs represent venous flow in vitro and do not account for the high shear rates and endothelial contribution of in vivo arterial flow. Thus, VHAs have not demonstrated sensitivity to von Willebrand factor activation which occurs with exposure to subendothelial collagen [78,79,80]. One recently described solution to account for these limitations of VHAs has been microfluidic channels lined with animal endothelium [81]. This ex vivo method may more accurately reflect vascular hemodynamics. Additionally, VHAs are not sensitive to Factor Xa inhibitors, direct thrombin inhibitors, or warfarin without the addition of specialized reagents [82,83,84]. Likewise, platelet receptor inhibition and antiplatelet medications are only detectable by specialized assays such as TEG PlateletMapping^®^ [85,86].

However, with the increased availability of highly reproducible POC devices such as the TEG^®^ 6 s, ROTEM^®^ Sigma, ClotPro^®^, and Haemosonics Quantra^®^, prior operator-dependent pipetting variability is being overcome. The recent US Food and Drug Administration approval of the cartridge system-based TEG^®^ 6 s specifically for trauma has also paved the way for prehospital use [87,88,89]. Specifically, the TEG^®^ 6 s is smaller, lightweight, portable with a handle, and has a readout that occurs directly on the device screen. Moreover, the TEG^®^ 6 s addresses the legacy TEG^®^ 5000′s need for expert operators, titration of reagents, and frequent recalibration [51,89,90,91].

## 4. TEG^®^/ROTEM^®^ in the Far-Forward and Austere Environment

Trauma in the far-forward and austere environment differs from urban trauma in several respects, the most significant of which is the difference in transport times. Urban civilian populations generally have access to prompt emergency medical services, whereas the conditions present in many military and civilian austere environments can often prevent transport for hours to days. This is what initially led to the use of WB in far-forward resuscitation strategies. However, as with urban settings, WB is given in these settings as a default in the initial hour of resuscitation. Subsequent treatment generally follows set blood product ratios coupled with CCA monitoring [92].

Military trauma specialists took note of their civilian colleagues’ use of VHAs and employed VHAs throughout the Iraq and Afghanistan conflicts to impressive effect (summarized in Table 1) [15,22,93,94,95]. For example, at the Bagram Airfield, Afghanistan, a significant improvement in adherence to DCR 1:1 FFP: RBC guidelines was observed after the deployment of ROTEM^®^ in the field [94]. Moreover, the ROTEM-guided group received 2 times and 4 times as many units of platelets and cryoprecipitate, respectively. No mortality benefit was detected between the two groups at 24 h or 30 d; however, the optimization of BCT therapy when guided by ROTEM^®^ did significantly decrease primary cause of death by exsanguination within the first 24 h (*p* < 0.03).

Additional studies have used ROTEM^®^ to evaluate the longitudinal effects of DCR on TIC in the combat setting [93]. One such study by Lammers et al. collected data from 3320 patients who received far-forward resuscitation at the US-led NATO Role III Multinational Medical Unit in Kandahar, Afghanistan between 2008 and 2016. Of those 3320 patients, 594 patients received goal-directed therapy that was guided by ROTEM^®^ [22]. The use of ROTEM^®^ allowed for more targeted treatments than those who received the standard fixed ratio of blood products. Specifically, targeted treatments involved less crystalloid and cryoprecipitate use during the first four hours of resuscitation. Additionally, the ROTEM^®^ cohort received more RBC units than the non-ROTEM^®^ cohort. Accordingly, the ROTEM^®^ cohort had a higher percentage of patients who met the definition for MT (more than 10 units of RBCs) compared to the non-ROTEM^®^ cohort. As a result, the ROTEM^®^ cohort also received more plasma-rich resuscitation. In the end, those 594 patients experienced significantly less coagulopathy, shock, and mortality than the non-ROTEM^®^ cohort [22].

Some results of Lammers’ study, such as the decreased use of crystalloid, were not altogether surprising. However, other results, such as the preferential use of plasma over cryoprecipitate and the number of patients requiring MTs, were unexpected. It has been noted that these and similar results show that not all hemorrhages can be treated the same, and a reliance on 1:1:1 paradigms may lead to preventable death [34].

The use of VHAs in combat and prehospital settings to help guide resuscitation has unique logistics challenges. The preparation, handling, storage, and in-flight transport of samples in special containers has also facilitated the widespread adoption of prehospital assessment of hemostatic integrity of patients in need of resuscitation for severe hemorrhage. In particular, the TEG^®^ 6 s has been tested in ground transport and during simulated and live aviation evacuation [51,96,97]. In these areas, newer generation VHAs (e.g., TEG^®^ 6 s, ROTEM^®^ sigma) correlate well with the traditional VHAs [88,90,97,98,99,100]. However, the correlation is not linear and is influenced by environmental factors such as temperature and barometric pressure as well as aeromedical-induced vibratory interference [51,97,99].

As a result, these preliminary experiences demonstrated that prehospital assessment of hemostatic integrity with VHAs is near at hand. For example, there has been a growth of POC devices which provide reliable and portable assays for blood chemistry, coagulation, and routine blood counts. Most recently, the importance of early detection of fibrinogen contribution and/or deficiency in the pathophysiology of early TIC has demonstrated that highly reproducible and portable assays for fibrinogen can be performed with 3–5 min and are available for use in-flight [56,101].

The use of WB was first championed in the far-forward combat area by Cannon in World War I, then in civilian austere environments, and now in urban environments. The use of VHAs during the first step of patient care has continued to expand, whether at the combat support hospital, the critical access hospital, or at the urban fixed trauma center.

## 5. Considerations for TEG^®^/ROTEM^®^ Platforms in the Field

### 5.1. Transport

Given that transport from the far-forward and austere environment can take considerable time, en route [or prehospital] care is arguably the next best setting to develop protocols for the use of VHAs. Prehospital VHAs in the far-forward combat and civilian austere environment may be useful to guide (1) initial resuscitation before transport, (2) resuscitation en route by air or ground transport, and (3) additional resuscitation on arrival to the combat support or rural hospital [102]. Expansion of VHA use in the prehospital setting as an adjunct to the administration of WB is necessarily limited by the reliability of TEG^®^/ROTEM^®^ under commonly encountered environmental insults. However, research into these unique transport challenges is as nascent as the concept of using VHAs in the far-forward venue (Table 2).

In 2019, Roberts et al. utilized a porcine trauma model to show that a TEG^®^ 6 s device operating during ground transport demonstrated results similar to those from a control, stationary TEG^®^ 5000 device [99]. However, a key difference between the transport from far-forward and austere environments versus urban settings is that the former are primarily air-based. Therefore, there has been recent interest in assessing the performance and fidelity of VHA platforms on rotary and fixed-wing aircraft. In a simulation trial with cartridge system TEG^®^ 6 s, reliability of the tests for R, K, α-angle, and MA were assessed during artificially generated vibration patterns that mimicked those experienced during helicopter take-off, flight, and landing. It was suggested that future non-simulation testing be conducted for the cartridge VHAs [96]. Additional testing in simulated [97], non-human [99], and human [102] rotary flight environments showed that TEG^®^ 6 s could not be relied upon as an in-flight diagnostic tool. However, Bates et al. suggested that samples could be drawn in-flight and used in the TEG^®^/ROTEM^®^ platform immediately on touch-down at the fixed trauma facility to guide continuing resuscitation efforts [102]. Yet, in-flight blood draws are impractical in many situations, indicating the need for innovative VHA technologies that use finger sticks. Additional prehospital studies are underway to assess in-flight use of VHAs.

Introducing hospital-based resuscitation tools into the prehospital environment to test their feasibility is not a new concept. Extracorporeal membrane oxygenation (ECMO) and mechanical ventilation have been deployed in aeromedical evacuation from far-forward combat support hospitals to advanced fixed facilities outside the military field (e.g., Landstuhl Regional Medical Center in Germany) [103]. However, there are often delays in this process, and many patients who go on ECMO in far-forward combat hospitals would benefit from earlier intervention [104]. This has led to recent calls for “ECMO packs” to be placed in combat support hospitals to improve survival. However, such packs would introduce further problems because ECMO patients are challenging to transport due to their hemodynamic instability and hypercoagulability induced by ECMO [105,106]. TEG^®^/ROTEM^®^ are frequently cited in ECMO algorithms as useful adjuncts in guiding therapy for the spectrum of coagulopathies induced by ECMO [107,108]. Therefore, widespread deployment of such ECMO packs may likewise need to be accompanied by far-forward TEG^®^/ROTEM^®^ kits to provide adequate information for hemostasis therapy guidance.

**Table 2 jcm-11-00356-t002:** Summary of literature investigating environmental influence on VHA accuracy.

Article	Participants	Type of Study and Setting	Conclusions
Cundrle et al., 2013 [109]	30 civilians treated with hypothermia for ROSC after cardiac arrest	Prospective Observational Setting: St. Anne’s University Hospital Brno, Czech Republic	Temperature adjustment for kaolin TEG^®^ or r-TEG^®^ are of little clinical utility due to low precision of TEG^®^ measurements; in vivo temperature TEG^®^ analysis is unnecessary.
Hunt et al., 2015 [61]	430 military and civilian (3 total studies)	Systematic Review and Meta-analysis	Due to insufficient studies, the authors found no evidence on accuracy of TEG^®^ and little evidence on accuracy of ROTEM^®^ to diagnose TIC when compared to PT/INR.
Jeppesen et al., 2016 [110]	40 civilians treated with hypothermia for ROSC after OHCA	Prospective Observational Setting: Aarhus University Hospital, Denmark	At 33 °C, ROTEM^®^ demonstrated a slower initiation of coagulation compared to 37 °C. The authors recommended that VHA analyses be maintained at 37 °C regardless of the patient’s body temperature.
Gill et al., 2017 [97]	One healthy volunteer	Comparative Methodological Analysis Setting: Sydney Children’s Hospitals Network, New South Wales, Australia	With the TEG^®^ 6 s, all measured parameters were significantly different while testing was subjected to motion.
Meledeo et al., 2018 [51]	3 healthy donors	Prospective Observational Setting: US Army Institute Surgical Research Blood Bank, San Antonio, Fort Sam Houston, Texas	TEG^®^ 6 s was more robust against motion and temperature stresses compared to the ROTEM^®^ delta and TEG^®^ 5000. TEG^®^ 6 s may be useful in austere environments.
Scott et al., 2018 [96]	148 TEG^®^ 6 s samples (72 AW139 Helicopter flight simulators with CAE 3000-series, 76 ground)	Comparative Analysis Setting: Toll ACE Training Centre, Bankstown Airport, Sydney, NSW, Australia	TEG^®^ 6 s was a reliable test in rotary wing flight conditions and demonstrated minimal variance compared to stable ground tests.
Roberts et al., 2019 [99]	8 swine on venovenous ECMO	Comparative Analysis Setting: San Antonio Military Medical Center, Fort Sam Houston, Texas	TEG^®^ 6 s during ground or aeromedical transport is feasible; however, method agreement was stronger at sea level and while stationary compared to mobile ground or altitude transport.
Bates et al., 2020 [102]	8 healthy donors	Prospective Observational Setting: Gold Coast University Hospital ICU, Gold Coast, Queensland, Australia; and in a LifeFlight Retrieval Medicine operated Leonardo AW139 Helicopter	ROTEM^®^ sigma and TEG^®^ 6 s were unreliable during flight, however remained calibrated post-flight and provided sound results over time.
Boyé et al., 2020 [45]	3 healthy donors 15 military ICU patients	Comparative Analysis Setting: ICU of the Military Medical Center Laveran (Marseille, France); simulated vibration at 100 Hz; simulated altitude of 8000 ft in a hypobaric chamber	TEG^®^ 6 s parameters at simulated 0 ft and 8000 ft were consistent for 9 of 13 parameters. TEG^®^ 6 s showed promise for aeromedical evacuation due to its ease of use and reliability.

INR, international normalized ratio; OHCA, out-of-hospital cardiac arrest; PT, prothrombin time; r-TEG^®^, rapid thromboelastography; ROSC, return of spontaneous circulation; ROTEM^®^, rotational thromboelastometry; TEG^®^, thromboelastometry; TIC, trauma-induced coagulopathy.

### 5.2. Altitude

Rapid changes in altitude can have profound effects on physiology in general and coagulopathy in particular. These changes can alter the baseline VHA results and consequently their interpretation. In an early study of 17 healthy volunteers who ascended from sea level to 5300 m, Martin et al. found that the TEG^®^ parameters R and k increased by 2.74 min (31%) and 2.59 min (107%), respectively; the *α* angle decreased by 6.1° (10%); and the maximum amplitude (MA) remained unchanged [111]. These changes are consistent with a hypocoagulable state, and a failure to account for these changes in either the far-forward or austere environment could lead to inappropriately aggressive or delayed treatment. The effects of full acclimatization on coagulation as measured by TEG^®^ were reported in 2018, by Rocke et al. [112]. In this study, 63 participants who resided at sea level ascended to 5200 m over 7 days. They initially became hypocoagulable, in agreement with Martin et al., but eventually they developed a hypercoagulable state, which agrees with epidemiological data on stroke risk at high altitudes [113]. Lastly, the TEG^®^/ROTEM^®^ platforms themselves are susceptible to errors at barometric extremes. In a 2020 report, Boye et al. found that a TEG^®^ 6 s placed in a hypobaric chamber set to simulate ~2500 m of altitude differed in output compared to a sea-level TEG^®^ 6 s in 4 of 13 parameters [45]. Taken together, such insights highlight the need for more research on the specific situational challenges of bringing TEG^®^/ROTEM^®^ platforms to the far-forward and austere environment. These data could make a notable impact on treatment strategies in such austere environments as the Himalayas, where amateur mountaineering trauma is on the rise, and in the Hindu Kush, a zone of potential conflict.

### 5.3. Hypothermia

Hypothermia is known to induce coagulopathy, and this can have a significant impact on VHA output and interpretation. Several studies have shown that TEG^®^ measurements are significantly different from expected values in patients being treated for cardiac arrest with therapeutic hypothermia [110,114]. Similar effects have also been reported in neonates treated for hypothermia [115]. While these studies were performed in the hospital setting in non-trauma patients, they have applications in austere settings as well. For example, the number of trauma patients that arrive at the hospital with hypothermia has been estimated at 23–67% [116,117]. Patients with hypothermia have the highest risk of morbidity and mortality due to coagulopathy. In such cases, TEG^®^/ROTEM^®^ assays can provide critical information on the unique contributions of trauma and hypothermia to the patient’s coagulopathy. However, some have raised the point that the VHA differences noted in a patient progressing from a hypothermic to normothermic state are smaller than those seen between repeat measurements of the same normothermic patient [109]. While this temperature discrepancy was measured with the legacy TEG^®^ 5000 platform, the early work of Cundrle Jr. et al. showed that further research is needed to optimize TEG^®^/ROTEM^®^ operating conditions and output interpretation for hypothermic patients.

### 5.4. Time to Actionable Information

For nearly a decade, the TEG^®^/ROTEM^®^ platforms have been known to produce results faster than standard coagulation panels [21]. However, the “standard” TEG^®^ utilizes kaolin as an activating reagent of the intrinsic pathway, and the full test requires 30 min to an hour to obtain the entire suite of outputs. This time is greatly reduced in the rapid-TEG^®^ (r-TEG^®^) protocol, which uses tissue factor and kaolin to activate the extrinsic pathway and thereby reduce the reaction time (R) [118]. r-TEG^®^ can produce all conventional TEG^®^ results within 15 min. Like all TEG^®^ assays, the r-TEG^®^ reagents are stable and stored at 2–8 °C (35.6–46.4 °F). Other groups have attempted to adapt predictive algorithms to obtain the key results from standard TEG^®^ assays based on early reaction conditions [119,120]. A more recent effort by Pressley et al. analyzed 873 r-TEG^®^ readings and developed an algorithm that could predict the need for transfusion of platelets or RBCs in 4 and 5 min, respectively [121]. This work makes the cheaper and more accessible standard TEG^®^ arguably on par with the r-TEG^®^ run time. However, there is still more to be done because Pressley and coworkers could not predict LY30 or the need for plasma infusion using this algorithm.

## 6. Future Direction

### 6.1. Towards a Common-Sense Approach to VHAs in the Far-Forward Setting

We were unable to find studies on the use of on-scene (pre-transport) VHAs in the far-forward or austere environment. Additionally, there are no standard recommendations for the immediate use of VHAs for patients given WB or BCT in the field, whether in the combat, austere, or urban environments. However, studies in the prehospital urban population are in process, and we expect those insights to be similarly adapted in the far-forward environment as they have been in the past. While individual circumstances may still present challenges that will require further study to mitigate, we broadly envision VHA utilization in a stepwise manner in both far-forward and the austere environments similar to that initially proposed by Bates et al. (Figure 2 and Figure 3) [102].

However, with extensive experience at the most highly developed, efficient, and mature evacuation systems in Afghanistan—where there was complete communications and air superiority—military transfusion specialists understand that the transmission of even basic information such as blood pressure and treatments given on scene be quite challenging due to a lack of a ubiquitous communication system, multiple radio relays, and difficult terrains. As such, the most reliable information about care in the far-forward may typically be found on casualty cards written with permanent marker pen [13,122]. Significant message degradation results over the multiple nodes of communication such that when the patient arrives to the Battalion Aid Station, other than the occasional vital signs or initial treatment on the casualty card, little information is available about prehospital course. For this reason, VHA use on the scene is not feasible in the far-forward combat setting at this time. At the Battalion Aid Station where resuscitation is continued, it is crucial that VHAs be used as guide blood products as it would be done at any civilian hospital regardless of size. The use of handheld VHAs, for which there are prototypes currently under investigation [123], may allow POC testing at the scene by special operations medics who may carry these tools in golden hour backpacks. For now, the initial use of VHAs in the far-forward combat setting would be limited to the Battalion Aid Station where the initial resuscitation occurs.

For example, we envision that at the initial site of trauma while emergency services are administered and the patient is prepped for transport, the blood collected on scene will be analyzed by r-TEG^®^ (enhanced with either tissue factor or a rapid analysis package). The information from this device will likely be completed before the transport is underway and will therefore guide the medics for both initial resuscitation during transport as well as the transport path. For example, if a patient’s VHA results showed that blood products were indicated but were not available on the transport, a diversion to a nearer facility may be more appropriate than to attempt the entire transport at once. Because VHA is unlikely to be adapted to air transport in the near-term (and there are likely many ground transport situations in far-forward and austere environments to which VHA is poorly adapted), VHA would next be used either at the next physical stopping point or at the final destination. Importantly, the blood sample can be run as soon as the transport is stationary, which allows for reassessment of the patient’s coagulopathy much faster than traditional testing. Should ECMO be indicated at this midway point, then VHAs also provide the most rapid way of assessing the patient’s ECMO-induced hypercoagulability during transport. Similarly, if a patient scheduled for aeromedial transport had a midpoint VHA indicating that a patient’s coagulopathy required more immediate attention, then the flight plan could be adjusted to obtain necessary blood products. Finally, it is important to note that this paradigm is possible even with an unaugmented VHA platform. The device can simply be left with far-forward personnel, who would then radio results to the aeromedical or ground transport.

### 6.2. Conclusions

VHA use in the far-forward and austere environment is probably feasible in the near future and may improve survival. However, it is clear from the literature review above that more work must be done before VHAs can be relied upon in these settings. The logistics of deploying VHAs on various platforms needs additional investigation. Similarly, it will be necessary to elucidate the appropriate corrections needed to interpret the VHA results obtained while in specific environmental conditions. Additional understanding of the coagulopathies induced by altitude-related hypoxia, ECMO, and trauma will augment the interpretation of VHA results. Because of the large number of unknowns, it is unlikely that the next steps will be high-powered randomized clinical trials. Instead, as with the lessons learned by pioneers from Larrey and Cannon to Holcomb and military trauma specialists in the modern era, advancements in the use of VHAs in the far-forward will come in the form of individual practitioners applying state-of-the-art medicine to individual patients according to each patient’s needs.

## Figures and Tables

**Figure 2 jcm-11-00356-f002:**
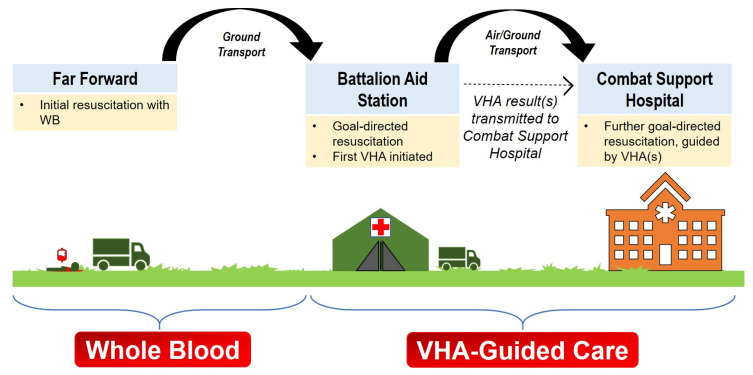
Proposed Use of VHA for Hemorrhagic Resuscitation in the Far-Forward Combat Environment. The hemorrhaging soldier is initially transfused WB (or fixed ratio) according to military protocol and blood product availability. While being resuscitated and prior to evacuation, the medics may obtain a blood sample and start the viscoelastic hemostatic assay (VHA). While the patient is en route to the Battalion Aid Station, the VHA result from the far-forward is transmitted to the Battalion Aid Station. Here, the far-forward VHA result—as well as a new VHA at the Battalion Aid Station—may guide goal-directed blood-component therapy. If the patient is to be transported further, the prior VHA results may be transmitted to the Combat Support Hospital. At the Combat Support Hospital, all future transfusions are guided by adjunctive VHA results to treat the patient’s individual hemostatic phenotype.

**Figure 3 jcm-11-00356-f003:**
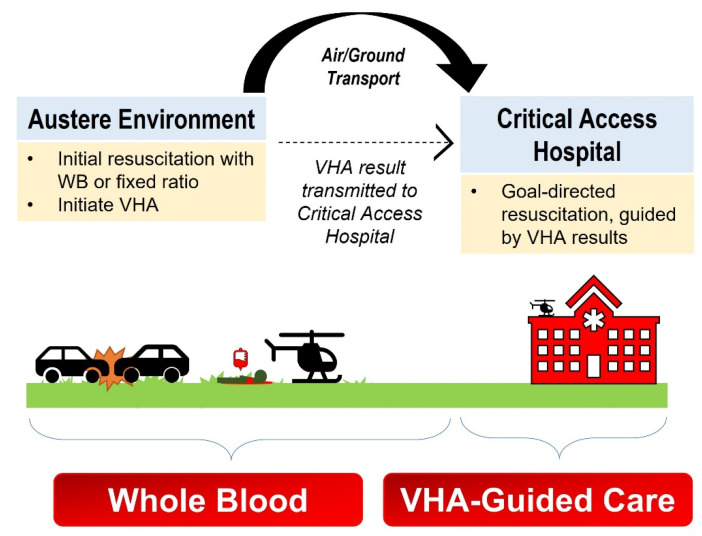
Proposed Use of VHA for Hemorrhagic Resuscitation in the Austere Civilian Environment. The hemorrhaging patient is initially resuscitated with WB or fixed ratio according to local protocol. First responder medics obtain a blood sample and begin running a viscoelastic hemostatic assay (VHA) at the site of injury, which likely will result while the patient is in transport. The VHA result is transmitted to the Critical Access Hospital. Here, the austere VHA and new VHA upon arrival to the Critical Access Hospital enables continued resuscitation comprising goal-directed blood-component therapy.

**Table 1 jcm-11-00356-t001:** Summary of military literature investigating VHA-guided resuscitation.

Article	Participants	Type of Study and Setting	Conclusions
Plotkin et al., 2008 [15]	44 military personnel with penetrating injuries	Retrospective Observational Setting: US Army Combat Support Hospital in Iraq	TEG^®^ as an adjunct to platelet counts and hematocrit was more predictive of blood transfusion than PT, aPTT, and INR together. Specifically, a reduced MA on TEG^®^ within 24 h of admission correlated with more administered blood products.
Doran et al., 2010 [93]	31 military personnel (19/31 received MT)	Prospective Observational Setting: United Kingdom Military; Camp Bastion, Helmand province, Afghanistan	ROTEM^®^ is feasible in the military setting and has a greater sensitivity for coagulation abnormalities compared to PT and aPTT.
Prat et al., 2017 [94]	219 military personnel (85 received ROTEM^®^-guided transfusion)	Retrospective Observational Setting: US Craig Theater Hospital, Bagram Airfield in Afghanistan	ROTEM^®^ did not significantly improve mortality or MT protocol activation. However, the ROTEM^®^-guided group received significant increases in PLT and CRYO transfusions (4× and 2×, respectively). ROTEM^®^ increased adherence to DCR protocol.
Cohen et al., 2019 [95]	40 military casualties	Prospective Observational Setting: NATO Hospitals in Afghanistan	ROTEM^®^ detected hemorrhagic coagulopathy and need for MT with greater sensitivity than INR alone. ROTEM^®^ should be included in MT protocols.
Lammers et al., 2020 [22]	3320 military personnel (594 received VHA-guided initial resuscitation)	Retrospective Observational Setting: US-led NATO Role III Multinational Medical Unit	VHA-guided resuscitation was independently associated with a decreased mortality (OR, 0.63; *p* = 0.001) and a 57% reduction in overall mortality (7.3% vs. 13.1%, *p* = 0.001).

aPTT, activated partial thromboplastin time; CRYO, cryoprecipitate; DCR, damage control resuscitation; INR, international normalized ratio; MA, maximum amplitude (TEG^®^ Parameter); MT, massive transfusion; PLT, platelet; PT, prothrombin time; ROTEM^®^, rotational thromboelastometry; TEG^®^, thromboelastography; VHA, viscoelastic hemostatic assay.

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
