# Peer review of "Hemorrhagic Resuscitation Guided by Viscoelastography in Far-Forward Combat and Austere Civilian Environments: Goal-Directed Whole-Blood and Blood-Component Therapy Far from the Trauma Center"

_jcm, 2022, doi:10.3390/jcm11020356_

Round 1
Reviewer 1 Report
Thank you for the privilege of reviewing this manuscript. The authors present a narrative review of the history and modern use of whole blood for military and austere civilian haemorrhagic trauma, and then further discuss the role of viscoelastic assays for these patients, and whether such assays can be pushed far forward in the battlespace.
Firstly, it must be said that this is a well written and expertly presented review, with a narrative thread that is both interesting and well referenced, with the flow and flare of experienced academic writers. It was a pleasure to review.
I only have a small number of minor comments, after which this article should be published:
- Most people are accustomed to referring to the Battle of Mogadishu as the “Black Hawk Down incident” but I think that term should be using inverted commas or italics (depending on journal style) since that is not its actual name, and made more popular by the books and films with that title. This applies to Figure 1 and page 3 line 106.
- The UK-led medical facility named Camp Bastion was in Helmand Province, whereas the US-led medical facility in Kandahar (at the Kandahar Air Field) was called the NATO Role III Multinational Medical Unit. Can the authors please have a look at page 7 lines 301-3 and Table 1 and correct?
- In Figure 3 the authors have used the term VET (I assume viscoelastic test) but they use VHA for the rest of the manuscript. Just needs changing to VHA
- The authors mention “surgeons” a few times as the primary decision-makers and custodians of trauma management (page 3, line 142; page 7, line 291; Conclusion line 508). I can understand this from an American standpoint where surgeons almost ubiquitously run trauma, but this is not necessarily always the case for international partners such as the UK and Europe, where anaesthetists and critical care physicians are equal partners. This is a minor point but worth considering because the authors cite some of the work from UK facilities, and also because this journal has an international readership.
- Subtitle 6.1 “in the Far forward” doesn’t quite work. Perhaps it should be “Far forward setting” or “Far forward environment”.
- How do the authors propose that the VHA results are transmitted along the chain of evacuation? Is this by radio? Or a print-out is sent with the casualty? Or does the machine get carried with the casualty along the chain? Having treated combat casualties at the point of injury, I know that even transmitting simple information such as blood pressure and treatments given can be challenging, usually using a combination of multiple radio relays, permanent marker pen and casualty cards. This often resulted in message degradation over the multiple nodes of communication so that the patient finally arrived in the Role 3 facility with barely any prehospital information of value. And this was in a mature, developed and highly efficient system in Afghanistan where we had complete communications and air superiority. Future conflicts are likely to be more challenging. Can the authors perhaps suggest something about the exact mechanism(s) proposed to make “front line” VHA work? This is absolutely critical for the whole premise!
- There are some authors listed as “funding acquisition” in the author contributions, but then the authors declare no funding in the funding statement. Not sure if this is a contradiction?
Once again, I congratulate the authors on an excellent piece of work. I look forward to seeing this in print.
Reviewer 2 Report
This is an interesting and complete narrative overview of blood resuscitation in austere and hostile environments. I think that the following arguments could be addressed in adjunct by authors:
- Which is the trigger to the use of WB in pre-hospital settings? Only the physiology of the patient or also the demonstration of a critical hemorrhage source? This question should be important particularly for blunt trauma where non compressible hemorrhage from internal organs are not easily detectable without imaging.
- A second concern is about the WB storage which is no longer than three weeks. Is there any possibility to use WB in hospital settings to avoid wasting of this important resource?
- Another concern is the use of VHA devices in austere environments and during transportation. At my knowledge there is no study which addresses this point. For all devices there is recommendation not to cause vibrations during use
Reviewer 3 Report
Firstly, I would like to congratulate the authors on this comprehensive yet clear and concise review article. The authors summarize the historic evolution of both fixed-ratio protocols and viscoelastically guided individualized treatment strategies for bleeding trauma patients. Following the description of their application in military and civilian settings, the authors outline the functional principles of the two most widely used devices, TEG and ROTEM. They then describe a number of specific challenges with regard to using TEG or ROTEM in extreme environments and conclude with their visions for both the practical use of viscoelastic guidance in extreme settings and potential future directions of studies in this field.
Overall, I would recommend the present article to be suitable for publication in Journal of Clinical Medicine. I have two minor comments/suggestions for the authors:
1.) page 5, 215-217: I would suggest to add traumatic brain injury here, e.g. doi: 10.1111/anae.14670.
2.) page 10, 383-397: Why do you add the topic of extracorporeal membrane oxygenation here? The only further mentioning of ECMO occurs in the conclusions on page 14, 505, where, however, the term "ECMO" does not add meaningful information to the sentence and could be simply skipped. Clearly, the intricate hemostatic alterations caused by ECMO therapy, not to speak of viscoelastic hemostatic guidance during ECMO support, are beyond the scope of this review article. To enhance focus and readability, I would therefore suggest to delete the paragraph on page 10, 383-397.
